# Evolution of SOMs' Structure and Learning Algorithm: From Visualization of High-Dimensional Data to Clustering of Complex Data

**Marian B. Gorzałczany** * 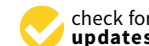 **and Filip Rudziński**

Department of Electrical and Computer Engineering, Kielce University of Technology, 25-314 Kielce, Poland; f.rudzinski@tu.kielce.pl

\* Correspondence: m.b.gorzalczany@tu.kielce.pl

**Abstract:** In this paper, we briefly present several modifications and generalizations of the concept of self-organizing neural networks—usually referred to as self-organizing maps (SOMs)—to illustrate their advantages in applications that range from high-dimensional data visualization to complex data clustering. Starting from conventional SOMs, Growing SOMs (GSOMs), Growing Grid Networks (GGNs), Incremental Grid Growing (IGG) approach, Growing Neural Gas (GNG) method as well as our two original solutions, i.e., Generalized SOMs with 1-Dimensional Neighborhood (GeSOMs with 1DN also referred to as Dynamic SOMs (DSOMs)) and Generalized SOMs with Tree-Like Structures (GeSOMs with T-LSs) are discussed. They are characterized in terms of (i) the modification mechanisms used, (ii) the range of network modifications introduced, (iii) the structure regularity, and (iv) the data-visualization/data-clustering effectiveness. The performance of particular solutions is illustrated and compared by means of selected data sets. We also show that the proposed original solutions, i.e., GeSOMs with 1DN (DSOMs) and GeSOMS with T-LSs outperform alternative approaches in various complex clustering tasks by providing up to 20% increase in the clustering accuracy. The contribution of this work is threefold. First, algorithm-oriented original computer-implementations of particular SOM's generalizations are developed. Second, their detailed simulation results are presented and discussed. Third, the advantages of our earlier-mentioned original solutions are demonstrated.

**Keywords:** artificial intelligence; computational intelligence; artificial neural networks; self-organizing neural networks; self-organizing maps; high-dimensional data visualization; complex data clustering

## 1. Introduction

Self-organizing neural networks—usually referred to as self-organizing maps (henceforward SOMs)—were introduced in the beginning of 1980s by T. Kohonen (see, e.g., [1,2]), who presented them as "a new, effective software tool for the visualization of high-dimensional data" (the quotation from Kohonen [1]). The visualization is performed by means of a topology-preserving mapping of the considered data into a low-dimensional display space (most often, in the form of a two-dimensional, usually rectangular, grid; three-dimensional SOMs—due to difficulties with their visualization—have achieved limited success [3]). In turn, according to [4], "feature mapping is conceptually different from clustering" and thus the authors of [4] conclude that SOM "is not a clustering method, but which often lends ideas to clustering algorithms" (see also a discussion in [5]). However, it is worth stressing that since the introduction of SOMs, their initial concept (including their structure and learning algorithm) has been significantly evolving and thus the range of its effective applications (including also complex clustering problems) has been substantially broadened. SOMs constitute an active research field, see,

e.g., "a varied collection of studies that testify to the vitality of the field of self-organizing systems for data analysis. Most of them relate to the core models in the field, namely self-organizing maps (SOMs)" (the quotation from Preface of the recently published [6]).

The objective of this paper is to briefly present several modifications and generalizations of the concept of SOMs (including our two original solutions)—starting from the simplest and ending with the most advanced ones—in order to illustrate their advantages in applications ranging from high-dimensional data visualization to complex data clustering. After brief presentation of a conventional SOM, two approaches that are able to automatically increase the number of neurons in their networks are outlined. They include Growing SOM (GSOM) [7] and Growing Grid Network (GGN) [8]. Next, a solution additionally equipped with the ability to add or remove some topological connections in the network is briefly presented. It is Incremental Grid Growing (IGG) approach [9]. In turn, Growing Neural Gas (GGN) approach [10] that can also remove some neurons from the network is outlined. Finally, our two original solutions that are able to automatically adjust the number of neurons in the network (to "grow"/reduce its structure), to disconnect it into substructures as well as to reconnect some of them again are outlined. They include Generalized SOMs with 1-Dimensional Neighborhood (GeSOMs with 1DN [11]) operating on splitting-merging neuron chains (they are also referred to as Dynamic SOMs (DSOMs) in [12,13]) and Generalized SOMs with splitting-merging Tree-Like Structures (GeSOMs with T-LSs [14–19]). The operation and performance of particular solutions are illustrated and compared by means of some data sets.

The contribution of this work is threefold. First, algorithm-oriented original computer-implementations of particular SOM's generalizations are developed. Second, their detailed simulation results are presented and discussed. Third, the advantages of our earlier-mentioned original solutions are demonstrated. Our simulation-based presentation of particular solutions is fully justified since—according to Kohonen himself [1] and Cottrell et al. [20]—"The SOM algorithm is very astonishing. On the one hand, it is very simple to write down and to simulate, its practical properties are clear and easy to observe. But, on the other hand, its theoretical properties still remain without proof in the general case, despite the great efforts of several authors" as well as "... the Kohonen algorithm is surprisingly resistant to a complete mathematical study. As far as we know, the only case where a complete analysis has been achieved is the one dimensional case (the input space has dimension 1) ...".

## 2. Conventional SOM, Its Selected Generalizations, and Related Work

In this section we characterize a conventional SOM and its earlier-mentioned generalizations. We also develop computer implementations of learning algorithms for particular SOMs' generalizations demonstrating, on the one hand, the evolution of the conventional-SOM learning technique and, on the other hand, the operation and performance of particular SOMs' models. A review of recent work is presented as well.

### 2.1. Conventional SOM

Consider a conventional SOM with most often used two-dimensional, rectangular map of neurons (with $m_1 \times m_2$ neurons) as shown in Figure 1 (the black lines represent physical connections in the network whereas the green lines—non-physical, topological ones). Assume that the map has $n$ inputs (features, attributes) $x_1, x_2, \ldots, x_n$ and consists of $m$ neurons ($m = m_1 \cdot m_2$); their outputs are $y_1, y_2, \ldots, y_m$, where $y_j = \sum_{i=1}^{n} w_{ji} x_i$, $j = 1, 2, \ldots, m$ and $w_{ji}$ are weights connecting the $i$-th input of the network with the output of the $j$-th neuron. Using vector notation ($x = [x_1, x_2, \ldots, x_n]^T$, $w_j = [w_{j1}, w_{j2}, \ldots, w_{jn}]^T$), $y_j = w_j^T x$. The learning data consists of $L$ input vectors $x_l$ ($l = 1, 2, \ldots, L$). Assuming that the learning vectors are normalized, the neuron $j_x$ which wins in the competition of neurons when the learning vector $x_l$ is presented to the network is selected in the following way:

$$d(\boldsymbol{x}_l, \boldsymbol{w}_{j_x}) = \min_{j=1,2,\dots,m} d(\boldsymbol{x}_l, \boldsymbol{w}_j), \tag{1}$$

where $d(\boldsymbol{x}_l, \boldsymbol{w}_j)$ is a distance measure between $\boldsymbol{x}_l$ and $\boldsymbol{w}_j$ (the most often used Euclidean measure is applied). A Winner-Takes-Most (WTM) algorithm is used in the learning process of the considered networks. The WTM learning rule is formulated as follows:

$$\boldsymbol{w}_j(k+1) = \boldsymbol{w}_j(k) + \eta_j(k)N(j, j_x, k)[\boldsymbol{x}(k) - \boldsymbol{w}_j(k)], \tag{2}$$

where $k$ is the iteration number, $\eta_j(k)$ is the learning coefficient, and $N(j, j_x, k)$ is the neighborhood function of the $j_x$-th winning neuron. Most often the Gaussian-type neighborhood functions are used, i.e.,:

$$N(j, j_x, k) = e^{-\frac{d_{tpl}^2(j, j_x)}{2\lambda^2(k)}}, \tag{3}$$

where $\lambda(k)$ is the neighborhood radius and $d_{tpl}(j, j_x)$—the topological distance between the $j_x$-th and $j$-th neurons. Figure 2 (Initialization-, WTM-algorithm-, and SOM-blocks) presents the block scheme of the learning algorithm of the conventional SOM. $e$ and $e_{max}$ in Figure 2 (and also in Figure 3) stand for epoch number and maximal number of epochs, respectively; the remaining symbols in both figures were introduced earlier in this section.

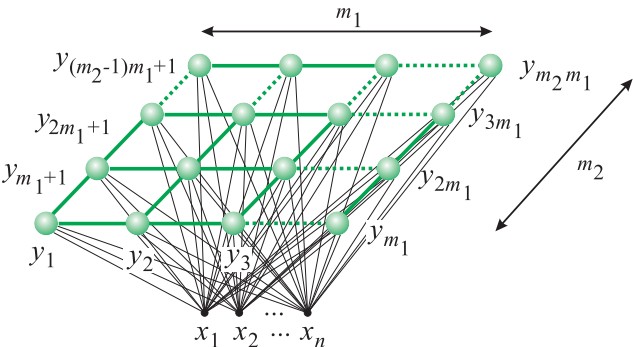

**Figure 1.** Conventional self-organizing map (SOM) with two-dimensional, rectangular map of neurons.

In order to visualize the distance- and density structures of the learning data on two-dimensional map of neurons, the so-called U-matrix has been proposed [5]. It consists of $m = m_1 \cdot m_2$ components $U_j$ ($j = 1, 2, \dots, m$) calculated as follows:

$$U_j = \sum_{h \in S_j} d(\boldsymbol{w}_h, \boldsymbol{w}_j), \tag{4}$$

where $S_j$ represents the neurons that are immediate neighbors of the considered $j$-th neuron. Therefore, a given $U_j$ represents the local distance structure in the input data. The U-matrix is usually visualized by means of shades in a gray scale (the smaller value of $U_j$—the lighter shade is used and vice versa, the larger $U_j$ corresponds to the darker shade).

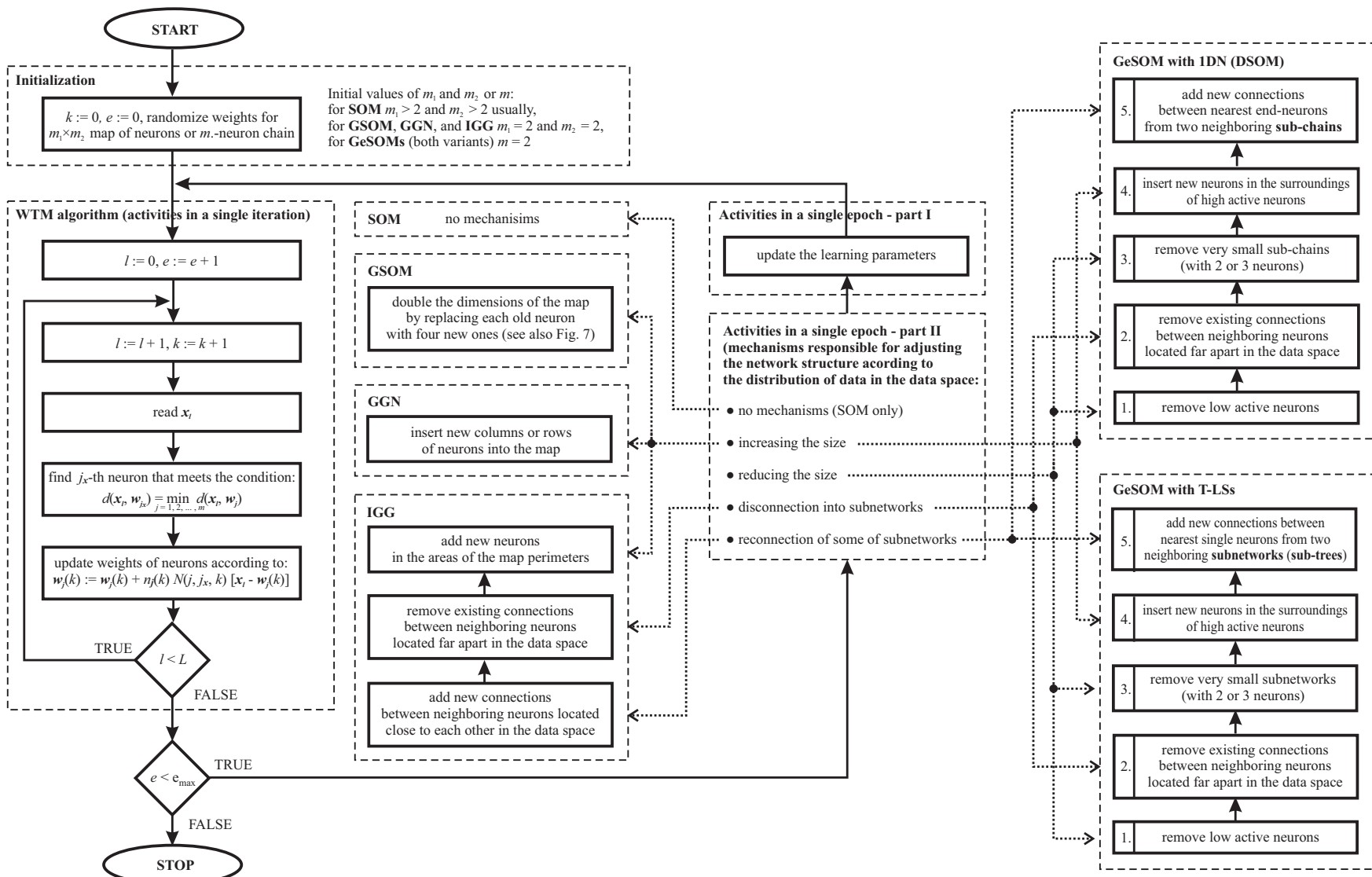

**Figure 2.** Block schemes of the learning algorithms of the conventional SOM, GSOM, GGN, IGG, GeSOM with 1DN (DSOM), and GeSOM with T-LSs.

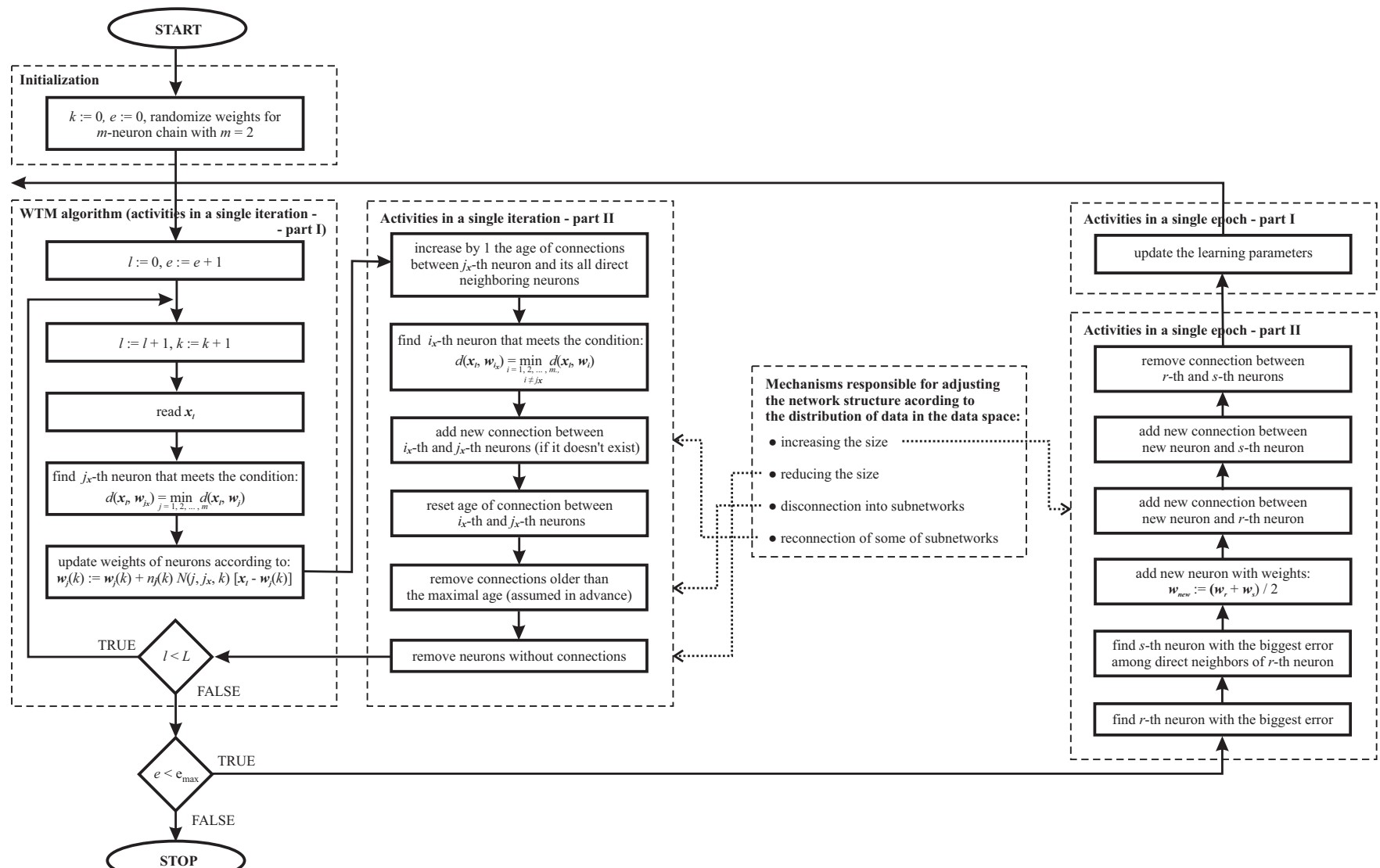

**Figure 3.** Block scheme of the learning algorithm of GNG.

First, we illustrate the operation of the conventional SOM in the case of the well-known and relatively simple *Zoo* data set available from the UCI repository of machine learning databases (https://archive.ics.uci.edu/ml). The *Zoo* data set contains 101 records; each record is described by 16 attributes (15 Boolean and 1 numerical). The number of classes (equal to 7) and the class assignments are known here; it allows us to directly verify the obtained results (obviously, the SOM—trained in an unsupervised way—does not utilize the knowledge of class assignments during the learning process).

Figure 4 presents the U-matrix (with $m_1 = m_2 = 30$) after performing the map calibration (see [1] for comments) for the considered data set. It is a two-dimensional display preserving, in general, the distance- and density structure of the 16-dimensional data set and showing, roughly, 7 data clusters (the darker areas of the U-matrix represent borders between them). The expressions $a(b/c)$ in Figure 4 should be read as follows: $a$ is the class label, $b$ is the number of $a$'s records assigned to $a$ in a given cluster, and $c$ is the overall number of $a$'s records (hence, the overall classification accuracy is equal to 95%).

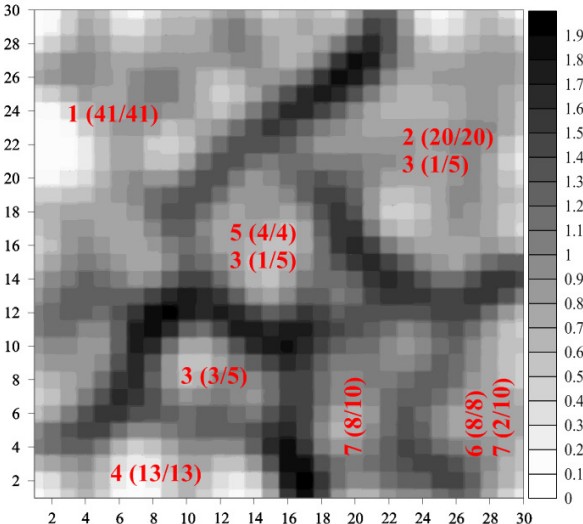

**Figure 4.** Gray-level U-matrix map for the visualization of the *Zoo* data set.

In turn, we consider two two-dimensional synthetic data sets characterized by different complexities and types of their data clusters. Obviously, two-dimensional data do not have to be visualized. However, they allow us to directly evaluate the performance of data visualization methods by comparing the visualization effects with original data. The first data set (see Figure 5a) contains five well-separated volume-type clusters whereas the second set (see Figure 6a) includes both thin, shell-like as well as volume-type clusters; their distribution and mutual relations are much more complex than in the first set. Figure 5b–i present the evolution (in selected learning epochs) of the conventional, two-dimensional SOM for the synthetic data set of Figure 5a. Figure 7 presents the U-matrix (with $m_1 = m_2 = 30$) for that data set. Analogous plots—for the synthetic data set of Figure 6a—are shown in Figure 6b–i and Figure 8.

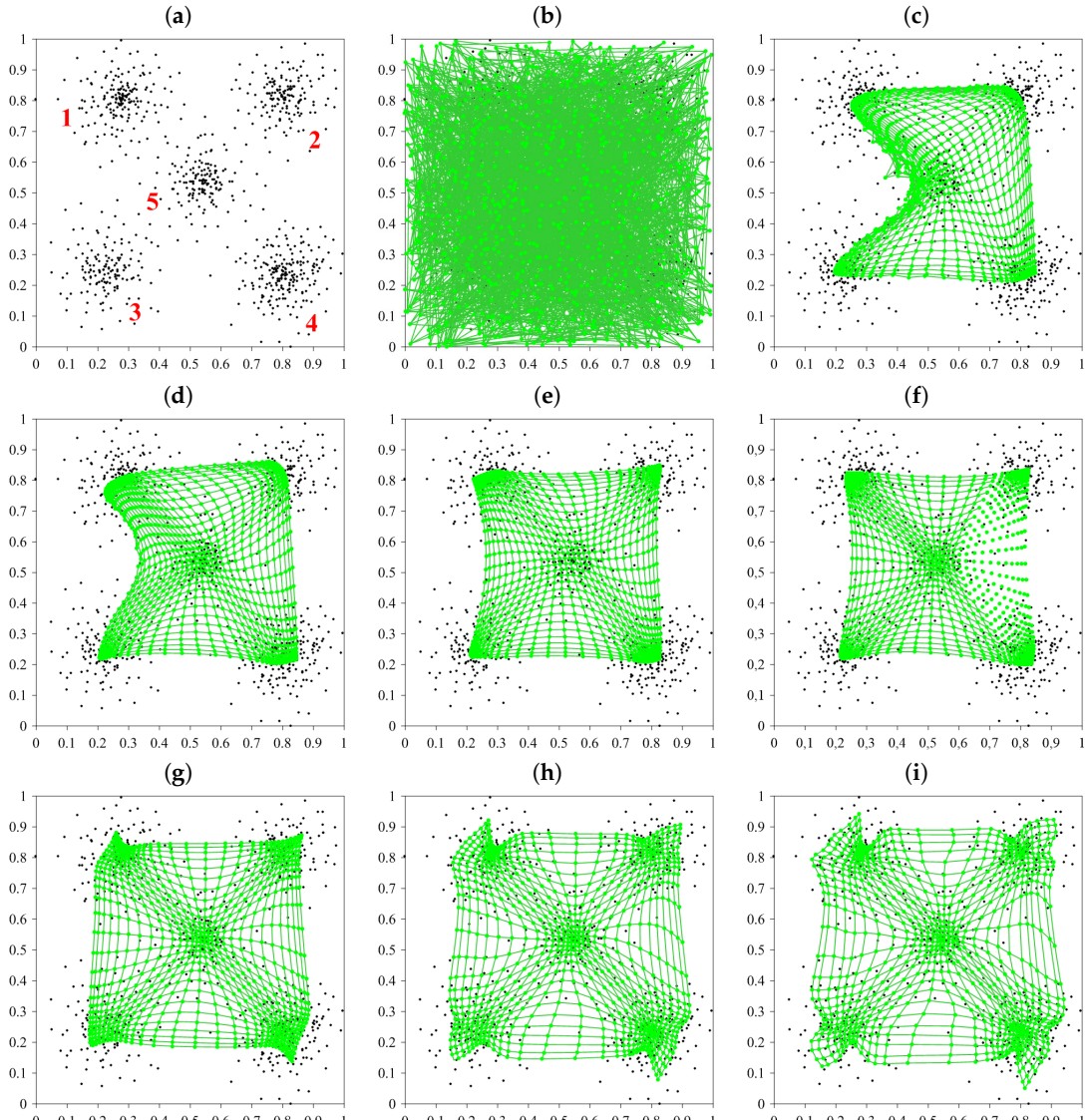

**Figure 5.** Synthetic data set (**a**) and the evolution of the conventional SOM in it in learning epochs: (**b**) No. 0 (start of learning), (**c**) No. 1, (**d**) No. 2, (**e**) No. 3, (**f**) No. 5, (**g**) No. 500, (**h**) No. 800, and (**i**) No. 1000 (end of learning).

For simple data set of Figure 5a, the conventional SOM generates clear and correct image of distance- and density structures in those data, detecting five distinct clusters as shown in Figure 7. However, for more complex data of Figure 6a, their image shown in Figure 8 is much less clear; the borders between some clusters are much less distinct (see, e.g., some regions between clusters 3 and 5 as well as 1 and 3). Moreover, Figsure 6c,d show that SOM has some problems in "unfolding" its structure at the initial stage of learning (or, equivalently, exhibits some tendency to "twist up" its structure) which may end up in wrong mapping of the learning data. It is caused by the processing of the whole map of $m_1 \times m_2$ neurons in each learning epoch (particularly, in the first phase of the map formation in which it should correctly "unfold" itself over the learning data). The processing of the whole map is also a very time-consuming operation. The introduction of Growing SOMs (GSOMs) [7] and Growing Grid Networks (GGNs) [8] is an attempt to address those problems.

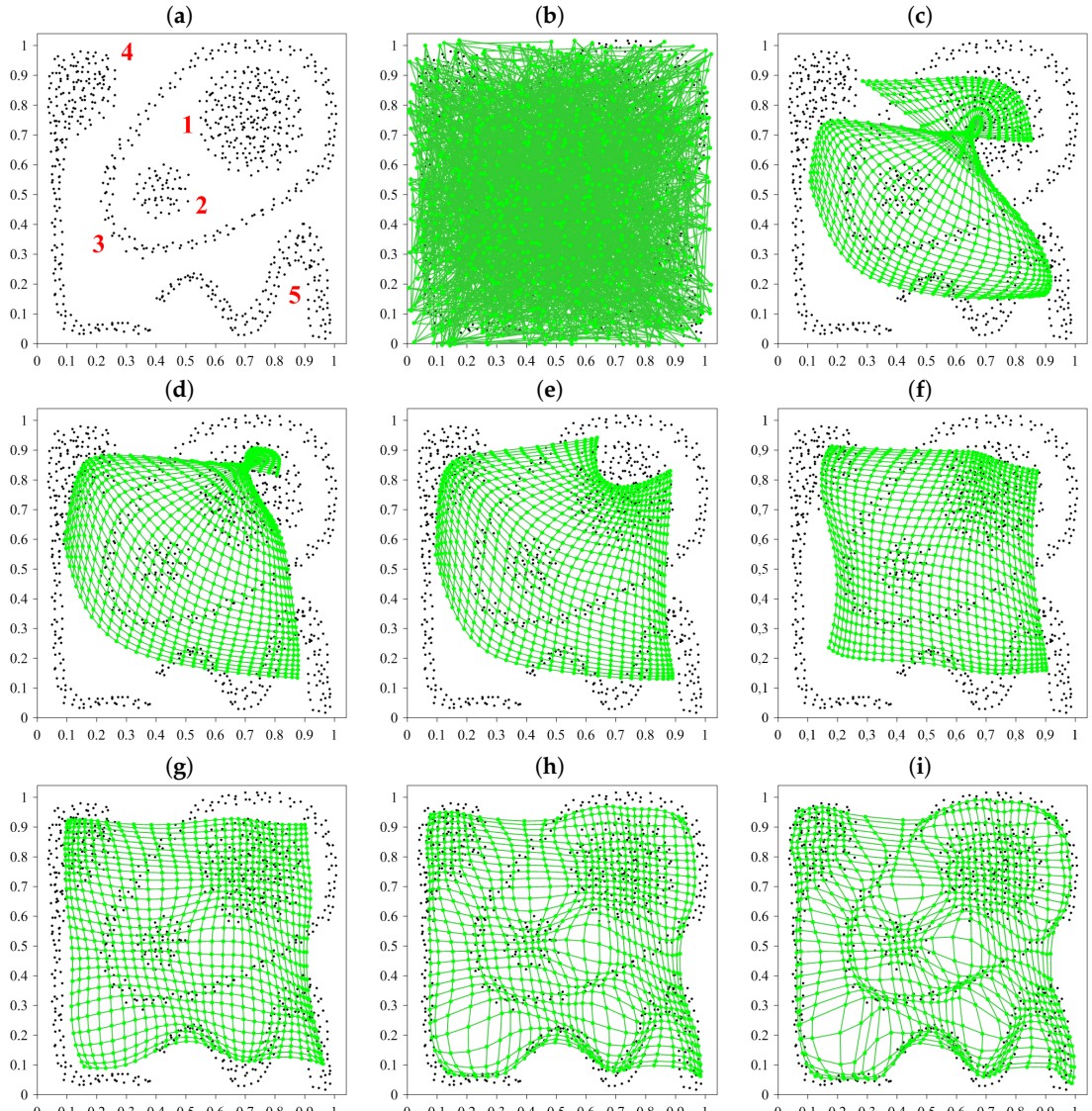

**Figure 6.** Synthetic data set (**a**) and the evolution of the conventional SOM in it in learning epochs: (**b**) No. 0 (start of learning), (**c**) No. 1, (**d**) No. 2, (**e**) No. 3, (**f**) No. 5, (**g**) No. 500, (**h**) No. 800, and (**i**) No. 1000 (end of learning).

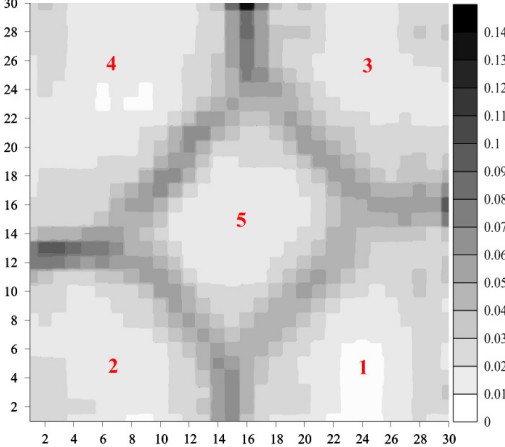

**Figure 7.** Gray-level U-matrix map for the visualization of the synthetic data set of Figure 5a (U-matrix is calculated for the neuron map of Figure 5i).

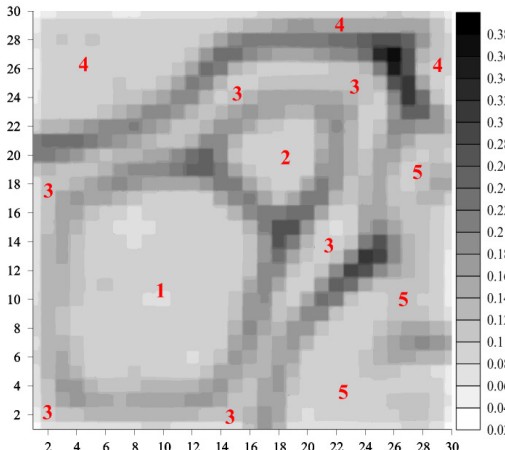

**Figure 8.** Gray-level U-matrix map for the visualization of the synthetic data set of Figure 6a (U-matrix is calculated for the neuron map of Figure 6i).

*2.2. Growing SOMs (GSOMs) and Growing Grid Networks (GGNs)*

GSOMs and GGNs are methods of improving the learning speed of the network and reducing the risk of poor "unfolding" its structure in the initial phase of the map formation. It is achieved by "starting the map with very few units and increasing that number progressively until the map reaches its final size" (a quotation from [7]). In GSOMs—according to [7]—"when the number of units increases, the locations of new units are interpolated from the locations of the old units"—see also the illustration in Figure 9. In turn, in GGNs—according to [8]—"by inserting complete rows or columns of units the grid may adapt its height/width ratio to the given pattern distribution". Figure 2 (Initialization-, WTM-algorithm-, and GSOM- or GGN-blocks) presents extensions of the conventional-SOM learning algorithm for the cases of GSOMs and GGNs. These extensions consist in modification of the Initialization block and adding the GSOM block or GGN block, respectively. As far as the initialization is concerned, the initial dimensions of the map are set to $m_1 \times m_2 = 2 \times 2$. The GSOM block introduces—at the end of each epoch—a mechanism that doubles the dimensions of the map by replacing each old neuron with four new ones as shown in Figure 9. In turn, the GGN block activates mechanism that inserts new columns or rows of neurons into the map.

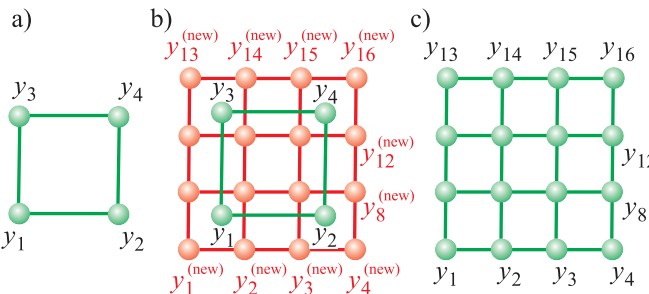

**Figure 9.** Increasing the Growing SOM (GSOM)'s structure: (**a**) before. (**b**) during, and (**c**) after performing a single step of the operation.

Figure 10 presents several stages of the GSOM's evolution for the more complex data set of Figure 6a (similar results are generated by GGN). In general, there are two stages in the map formation. In the first one the map "unfolds" itself to place the neurons in the desired order (see Figure 10a–g). In the second one the collection of neurons asymptotically approaches the distribution of data (see Figure 10h,i). Figure 11 shows the U-matrix (with final values of $m_1 = 32$ and $m_2 = 32$) for that data set. It is clear that GSOM is much more efficient than the conventional SOM in the initial phase of the map formation. However, the final results of both approaches are comparable (compare Figure 11 with Figure 8), i.e., the images of the input data in terms of detecting data concentration are

not very clear in some parts of both maps. Therefore, in the next stage, more advanced approaches using non-fully connected network topology should be considered to overcome this problem.

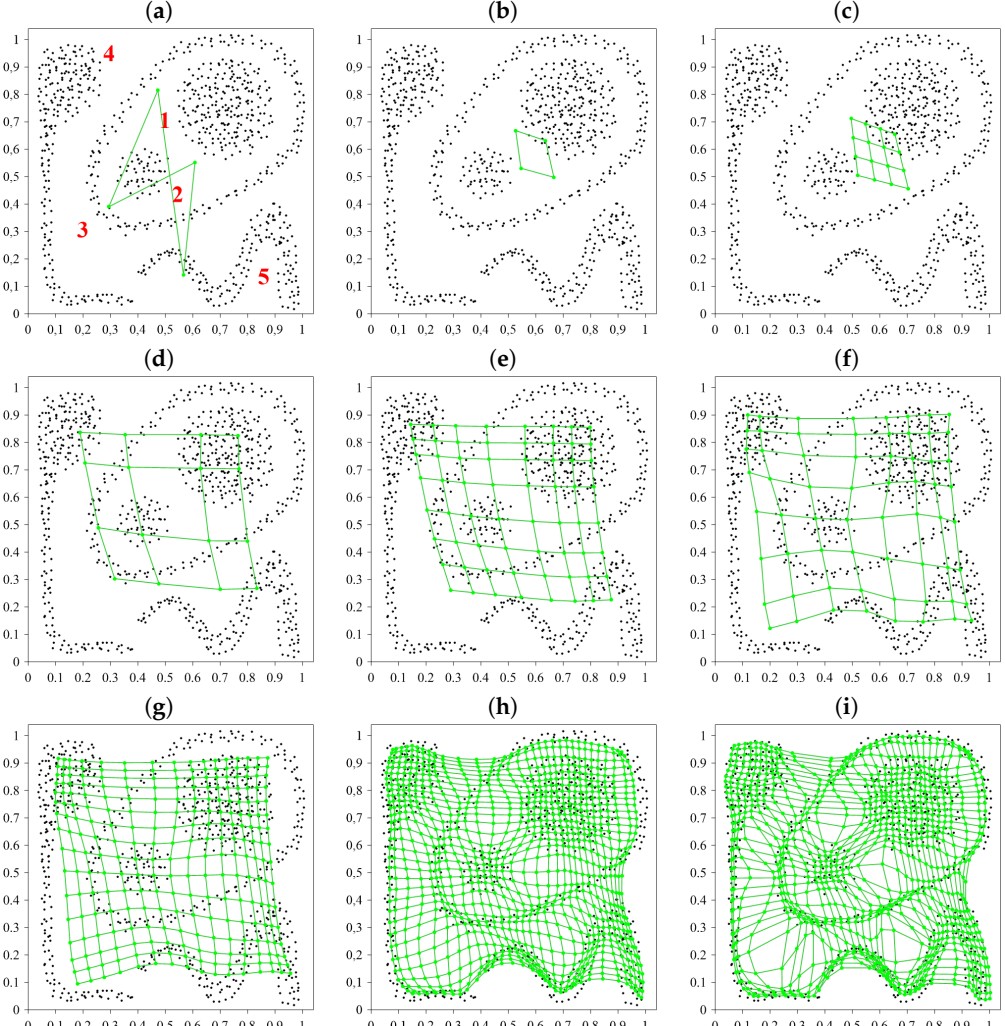

**Figure 10.** The evolution of GSOM in the synthetic data set of Figure 6a in learning epochs: (**a**) No. 0 (start of learning), (**b**) No. 1, (**c**) No. 2, (**d**) No. 3, (**e**) No. 4, (**f**) No. 5, (**g**) No. 6, (**h**) No. 10, and (**i**) No. 1000 (end of learning).

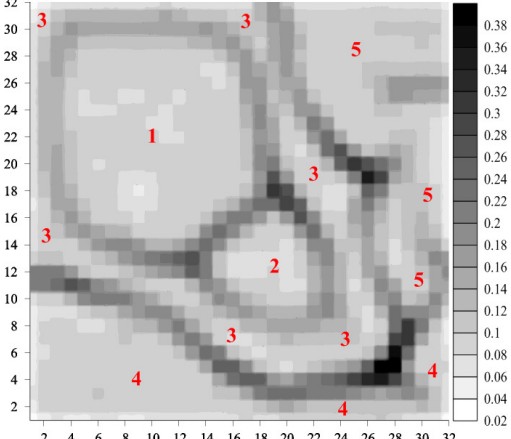

**Figure 11.** Gray-level U-matrix map for the visualization of the synthetic data set of Figure 6a (U-matrix is calculated for the neuron map of Figure 10i).

### 2.3. Incremental Grid Growing (IGG) Approach

IGG method is able to increase the number of neurons in the map (however, only in the areas of the map perimeter and not "inside" the map) as well as to remove or add some topological connections in the map [9]. Figure 2 (Initialization-, WTM-algorithm-, and IGG-blocks) presents further extension of the conventional-SOM learning algorithm for IGG approach. This extension consists in modification of the Initialization block (in a similar way as for GSOMs and GGNs, i.e., by setting the initial dimensions of the map to $m_1 \times m_2 = 2 \times 2$) and adding the three-module IGG block, which is activated after each learning epoch. The first module of the IGG block controls the process of adding new neurons into the area of the map perimeter. The second module governs the removal of existing connections between neighboring neurons located far apart in the data space. The third module manages the adding of new connections between neighboring neurons located close to each other in the data space.

The main idea behind the IGG approach is to build and organize the map in an incremental way. Such an approach initially organizes a small number of neurons in the structure. Next, after each learning epoch, the Euclidean distance between unconnected neighboring neurons in the map is examined. If the distance is below the "connect"-control parameter, a connection between the neurons is introduced. In an analogous way, using the "disconnect"-control parameter, existing connections between neighboring neurons located far apart in the data space are removed. It is also possible to add some neurons at the perimeter areas that inadequately represent the data.

Figures 12 and 13 illustrate the application of IGG approach to the synthetic data sets of Figures 5a and 6a, respectively. The evolution of the number of IGG's substructures as the learning progresses is shown in Figures 12j and 13j for particular data sets. The number of substructures is equal to the number of detected clusters. Similarly as for GSOMs and GGNs, there are two stages in the map formation. In the first stage, the map concentrates on "unfolding" its structure by an intensive increase of the number of neurons (see Figures 12k and 13k) without disconnecting its structure into substructures (see Figures 12j and 13j). In turn, in the second stage, the map with a relatively stable number of neurons concentrates on its essential task, i.e., detecting clusters in data by disconnecting its structure into substructures representing particular clusters. It is evident that IGG much better copes with simple data of Figure 5a, finally dividing its structure into six substructures (including one very small) representing five clusters. However, IGG completely fails in the case of more complex data of Figure 6a. It is clear that IGG lacks the ability to reduce the number of neurons in the network and to reconnect some substructures representing different part of a given cluster. It results in an "explosion" of separate, small substructures in the final part of the learning process (see Figure 13j) and completely false image of the cluster distribution in the considered data set.

### 2.4. Growing Neural Gas (GNG) Approach

GNG method—in comparison with IGG approach—can also remove some neurons from its structure. It is also able—under some circumstances—to reconnect some of its substructures [10]. GNG method combines the growing mechanism inherited from an approach referred to as Growing Cell Structures [21] with topology formation rules using the Competitive Hebbian Learning scheme [22,23]. The GNG approach starts with a few neurons (usually two) and new neurons are successively inserted (after a predefined number of learning epochs) in the neighborhood of the neurons featuring with the largest local accumulated error measure. The GNG method can also detect the inactive neurons (i.e., the neurons, which do not win during a long time interval) by tracing the changes of an "age" parameter associated with each connection in the network. Therefore, it is able to modify the network topology by: (i) removing connections with their "age" variable not being refreshed for a given time interval and (ii) removing the resultant inactive prototypes. We developed original computer implementation of the GNG learning algorithm—its detailed block scheme is shown in Figure 3.

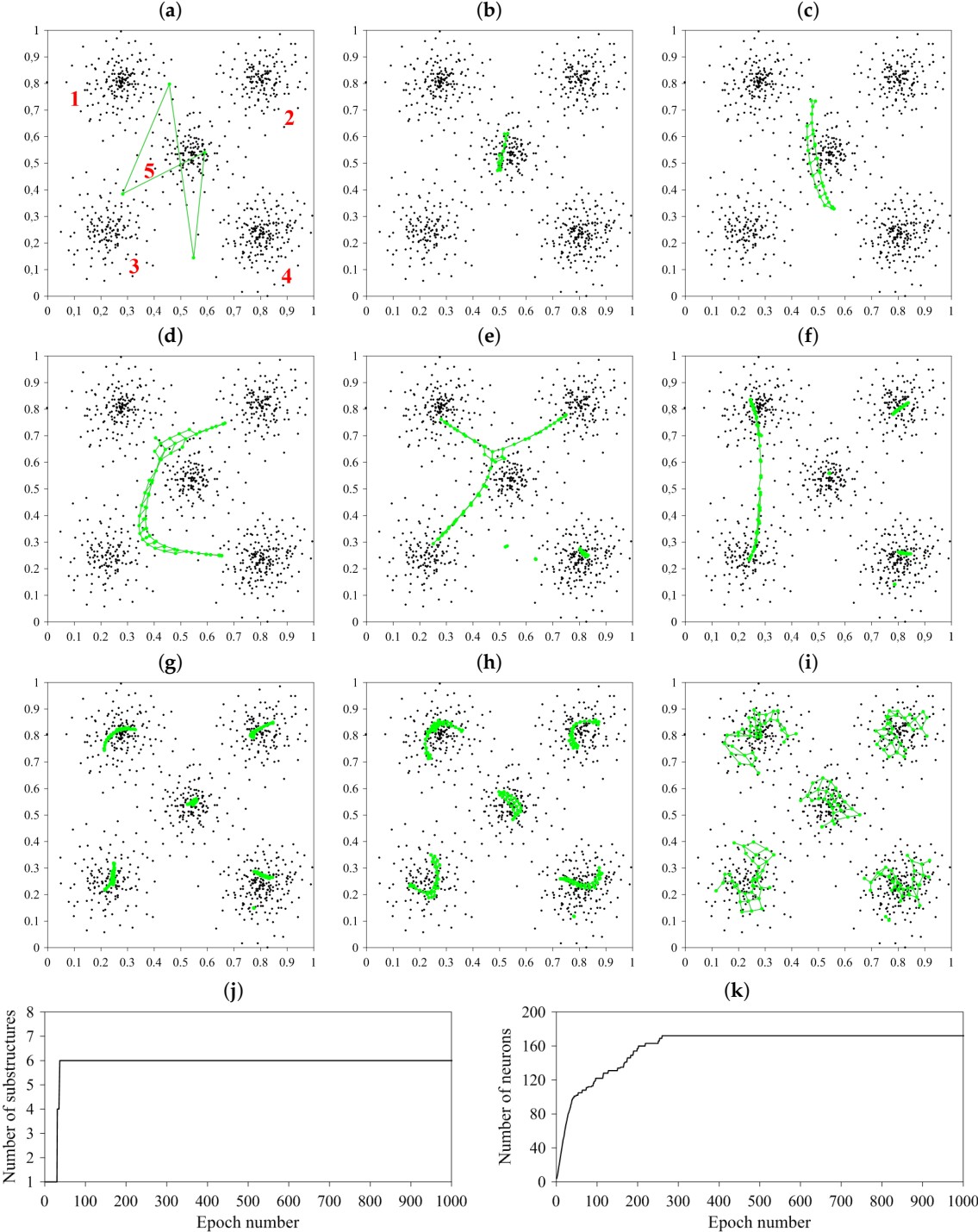

**Figure 12.** The evolution of Incremental Grid Growing (IGG) in the synthetic data set of Figure 5a in learning epochs: (**a**) No. 0 (start of learning), (**b**) No. 5, (**c**) No. 10, (**d**) No. 20, (**e**) No. 30, (**f**) No. 50, (**g**) No. 100, (**h**) No. 500, and (**i**) No. 1000 (end of learning) as well as plots of the number of substructures (**j**) and the number of neurons (**k**) vs. epoch number.

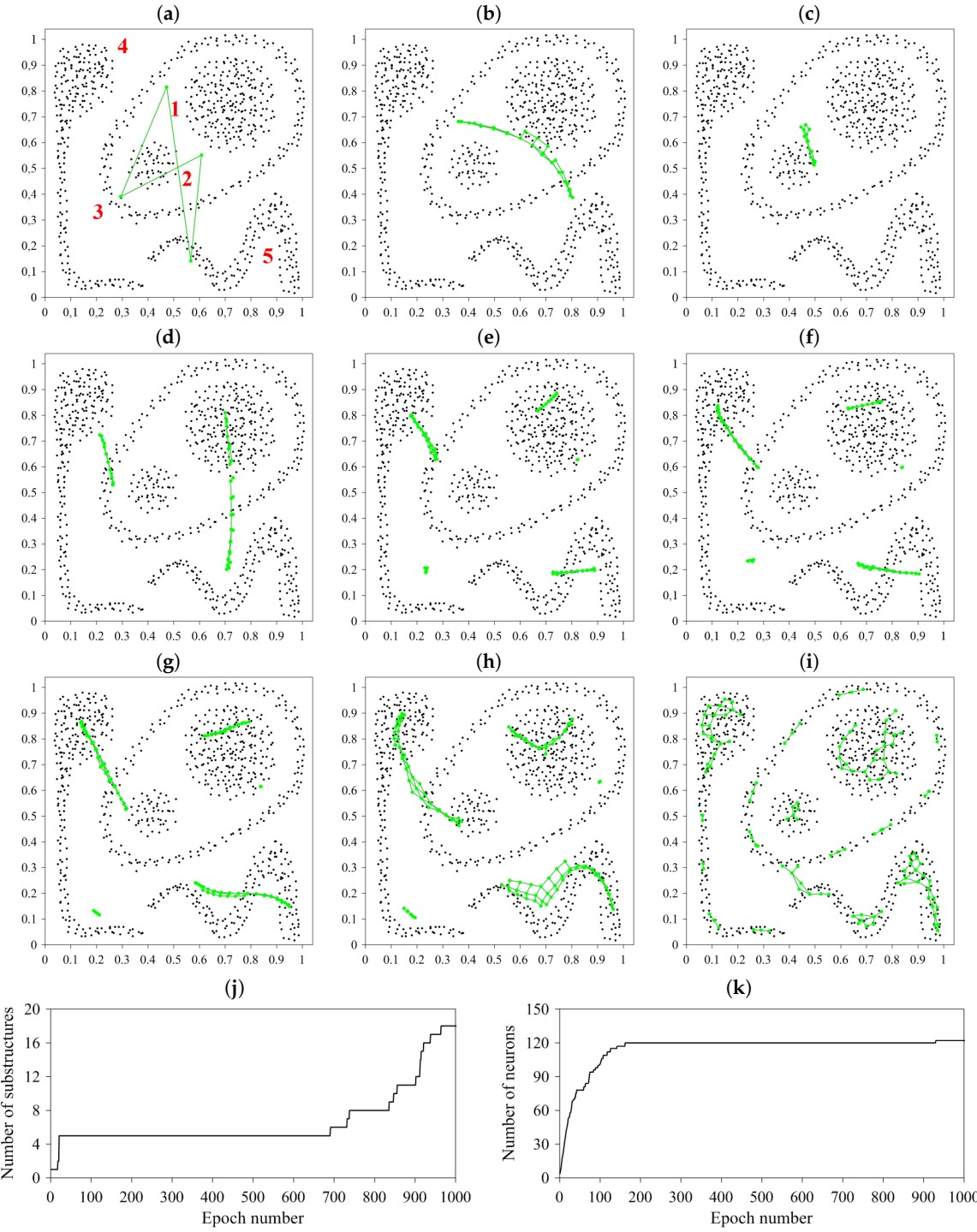

**Figure 13.** The evolution of IGG in the synthetic data set of Figure 6a in learning epochs: (**a**) No. 0 (start of learning), (**b**) No. 5, (**c**) No. 10, (**d**) No. 20, (**e**) No. 30, (**f**) No. 50, (**g**) No. 100, (**h**) No. 500, and (**i**) No. 1000 (end of learning) as well as plots of the number of substructures (**j**) and the number of neurons (**k**) vs. epoch number.

Figure 14 illustrates the application of GNG method to more complex data set of Figure 6a. Unfortunately, this time the "unfolding" stage of the network takes about 700 (out of 1000) learning epochs. During the first part of that stage (see Figure 14k), the network concentrates on increasing its number of neurons and spreading them in the data space. Discovering clusters by disconnecting GNG's structure into substructures (see Figure 14j) is performed during the last 300 learning epochs.

But on the positive side, the clustering results of GNG are much better than those of IGG. GNG detects correctly four clusters out of five. An incorrect decision regards only clusters Nos. 4 and 5; their "thin ends" are close to each other which results in combining both clusters into one entity.

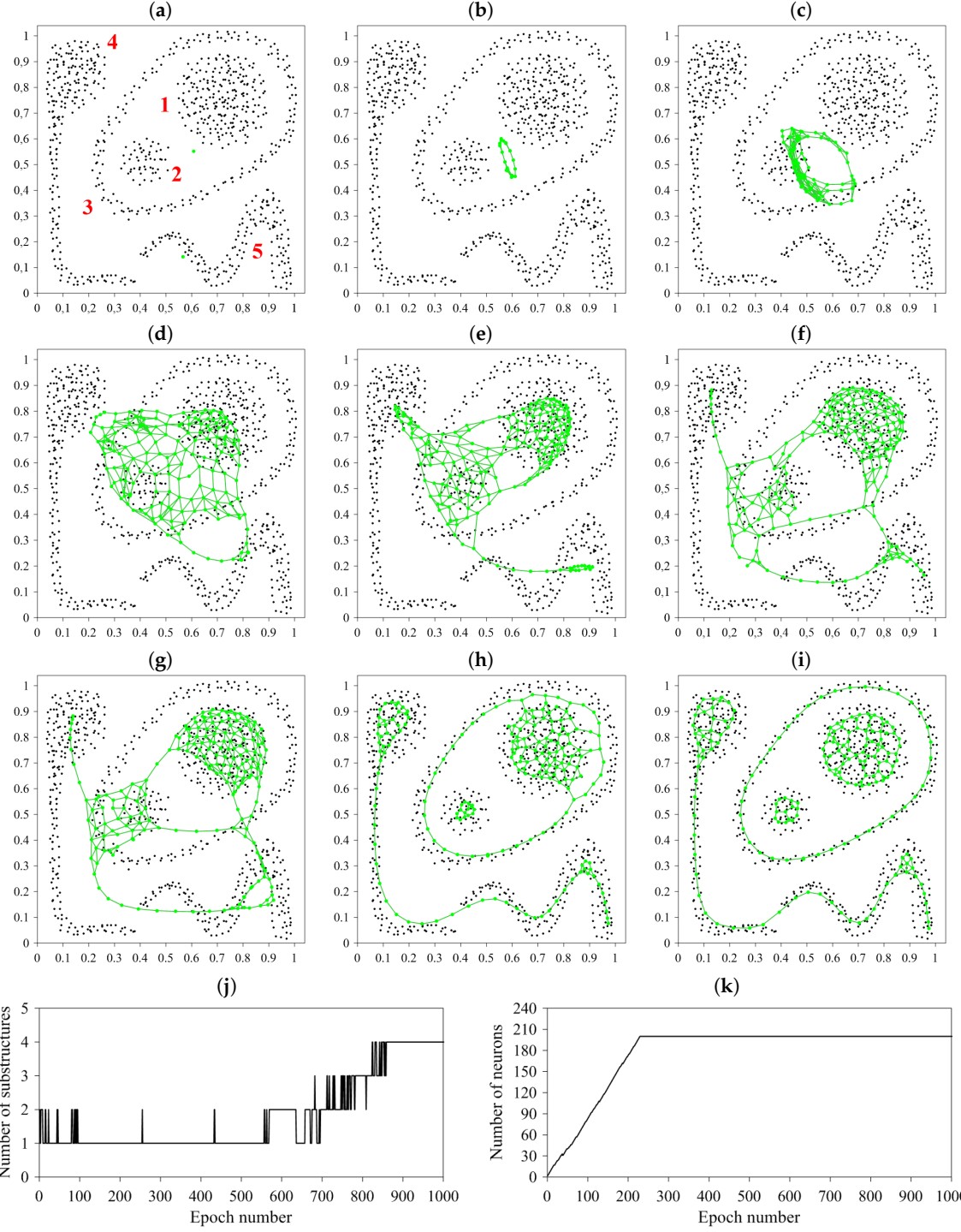

**Figure 14.** The evolution of Growing Neural Gas (GNG) in the synthetic data set of Figure 6a in learning epochs: (**a**) No. 0 (start of learning), (**b**) No. 5, (**c**) No. 10, (**d**) No. 20, (**e**) No. 30, (**f**) No. 50, (**g**) No. 100, (**h**) No. 800, and (**i**) No. 1000 (end of learning) as well as plots of the number of substructures (**j**) and the number of neurons (**k**) vs. epoch number.

*2.5. Related Recent Work*

Earlier in this section we presented several SOM's generalizations illustrating the evolution of their applications that range from data visualization to data clustering. We also developed computer implementations of learning algorithms for particular approaches and we showed simulation results illustrating their operation. In this section we present some literature context for the SOMs-based data visualization and clustering by briefly reviewing recently published related work.

An extension of U-matrix Equation (4) for data visualization is proposed in [24]. However, although the generalized U-matrix is able to visualize the similarities and dissimilarities among high-dimensional data points in a scatter plot of the projected points, it is unable to visualize the disruption of clusters, based on which the quality of structure preservation is defined [25]. An extension of the SOM that utilizes several landmarks, e.g., pairs of nodes and data points (referred to as LAMA) is proposed in [26]. LAMA generates a user-intended nonlinear projection in order to achieve, e.g., the landmark-oriented data visualization. However, missetting of the LAMA's learning parameters, which are manually adjusted, may cause the mesh grid of codebook vectors to be twisted or wrinkled [26]. An emergent SOM (ESOM) concept is proposed in [25]. The structure of a ESOM's feature map is toroidal, i.e., the borders of the map are cyclically connected. It allows the problem of neurons on map borders and thus, boundary effects to be avoided. However, the structure of the input data emerges only when U-matrix SOM-visualization technique is used [25]. In turn, a nonlinear dimensionality reduction method for data visualization based on combination of extreme learning machine (ELM) approach with SOM and referred to as ELM-SOM+ is proposed in [27]. The SOM-based data visualization finds diverse applications ranging from gearbox fault feature extraction and classification [28] to evaluation of ground water potential to pollution in Iran [29]. A solution building a bridge between SOM-based data visualization and clustering is an improved SOM network combining clustering and clustering visualization into the same computational process. Such an approach—applied to gene data cluster analysis—is proposed in [30].

As far as SOM-based or SOM-inspired data clustering is concerned, a SOM prototype-based methodology for cluster analysis is proposed in [31]. It makes use of topology preserving indices by means of which the most appropriate size of the map is selected. It also automatically analyzes the most relevant number of clusters in data. However, one real-life example in [31] does not allow for a deeper evaluation of its performance. Hierarchical SOM-based approach that can handle numerical, categorical, and mixed data clustering is proposed in [32]. In turn, a SOM-inspired algorithm for partition clustering is proposed in [33]. The SOM-based or inspired data clustering finds applications ranging from dengue expression data analysis [33] to house data grouping [34]. It is worth mentioning here also a collection of earlier published two-level approaches to data clustering using SOMs [35–38] (see also [39,40] for brief reviews). At the first level, each data point is mapped, by the standard SOM approach, into a lower dimensional space that preserves the data topology. Then, at the second level, the obtained data are subject to clustering using traditional methods.

## 3. The Proposed Generalized SOMs with Splitting-Merging Structures

Finally, we would like to briefly present our two original approaches, i.e., (i) Generalized SOMs with 1-Dimensional Neighborhood (GeSOMs with 1DN [11] also referred to as Dynamic SOMs (DSOMs) in [12,13]) which operate on splitting-merging neuron chains (see Figure 15) and (ii) Generalized SOMs with splitting-merging Tree-Like Structures (GeSOMs with T-LSs [14–19]; see Figure 16). Main features, main objectives, general concept, and implementation of both approaches are the following:

*Main features*: Our approaches work in a fully unsupervised way, i.e., (i) they do not need to predefine the number of clusters and (ii) they use unlabeled data.
*Main objectives*: (i) an automatic determination of the number of clusters in a given data set and (ii) an automatic generation of multi-point prototypes for particular clusters.

*General concept*: In comparison with conventional SOMs, the proposed approaches are also equipped with three additional mechanisms (allowing for data segmentation) such as: (i) automatic adjustment of the number of neurons in the network (removing low-active neurons and adding new neurons in the areas of the existing high-active neurons), (ii) automatic disconnection of the tree-like structure into subnetworks, and (iii) automatic reconnection of some of the subnetworks preserving—in the case of GeSOMs with T-LSs—the no-loop spanning-tree properties. Such generalized SOMs are able to detect data clusters of virtually any shape and density (including both volumetric and thin, shell-type clusters) by locating a single disconnected subnetwork in the area of the data space occupied by a given cluster. Hence, the number of automatically generated subnetworks is equal to the number of clusters. Moreover, the set of neurons in a given subnetwork is a multi-prototype of the corresponding cluster. Such prototypes can be directly used in clustering/classification tasks using the well-known nearest multi-prototype algorithms [41].

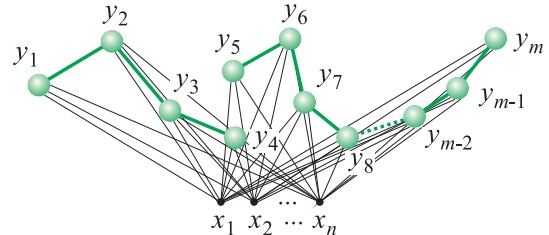

**Figure 15.** Generalized SOMs (GeSOM) with 1DN (DSOM).

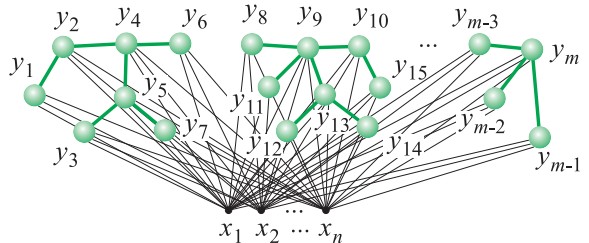

**Figure 16.** GeSOM with T-LSs.

*Implementation:* Figure 2 (Initialization- and WTM-algorithm-blocks as well as blocks concerning GeSOM with 1DN and GeSOM with T-LSs) presents generalizations of the conventional-SOM learning algorithm for both structures considered. Both approaches start with a few neurons (usually two—see the Initialization block of Figure 2). In turn, after each learning epoch, the mechanism (i), (ii), and (iii) listed in the *General-concept* part of this section are implemented by conditional activation of five successive operations (see also Figure 2; for details, see the afore-listed references):

1.   The removal of single, low-active neurons preserving the network continuity: a given neuron is removed if its activity—measured by the number of its wins—is below an assumed level.
2.   The disconnection of the network (subnetwork) into two subnetworks: the disconnection of two neighboring neurons takes place if the Euclidean distance between them exceeds an assumed level.
3.   The removal of very small subnetworks with two or three neurons (usually, they represent noise).
4.   The insertion of additional neurons into the neighborhood of high-active neurons in order to take over some of their activities (it results in distributing more evenly the system's activity across the network).
5.   The reconnection of two selected subnetworks:

    5.1.   The GeSOM with 1DN case: the nearest end-neurons from two neighboring sub-chains are connected if the Euclidean distance between them is below an assumed level.

5.2. The GeSOM with T-LSs case: the nearest single neurons from two neighboring subnetworks are connected if the Euclidean distance between them is below an assumed level (this mechanism supports growing tree-like structure of the network).

Figure 17a–i present the selected stages of the evolution of GeSOM with 1DN (DSOM) for the more complex data set of Figure 6a. Figure 17j,k illustrate the adjustment of the number of sub-chains (finally equal to 5, i.e., the number of clusters) and the number of neurons (finally equal to 243) in the network as the learning progresses. Figure 17l shows the envelope of the distance histogram $H_j^{dist}$ between two neighboring neurons Nos. $j$ and $j+1$ along the neuron chain ($j = 1, 2, \ldots, m-1$):

$$H_j^{dist} = d(\boldsymbol{w}_j, \boldsymbol{w}_{j+1}). \tag{5}$$

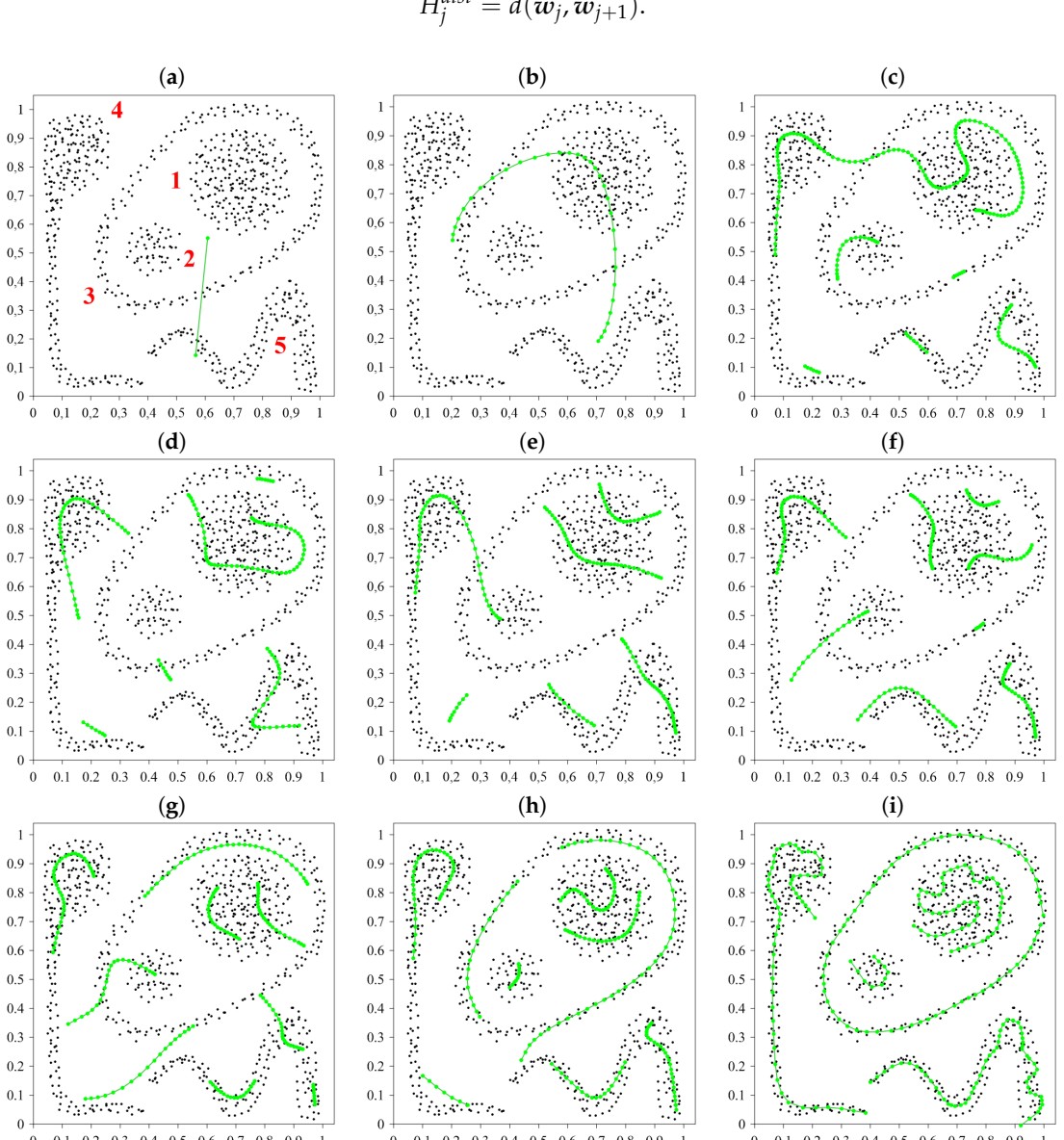

**Figure 17.** *Cont.*

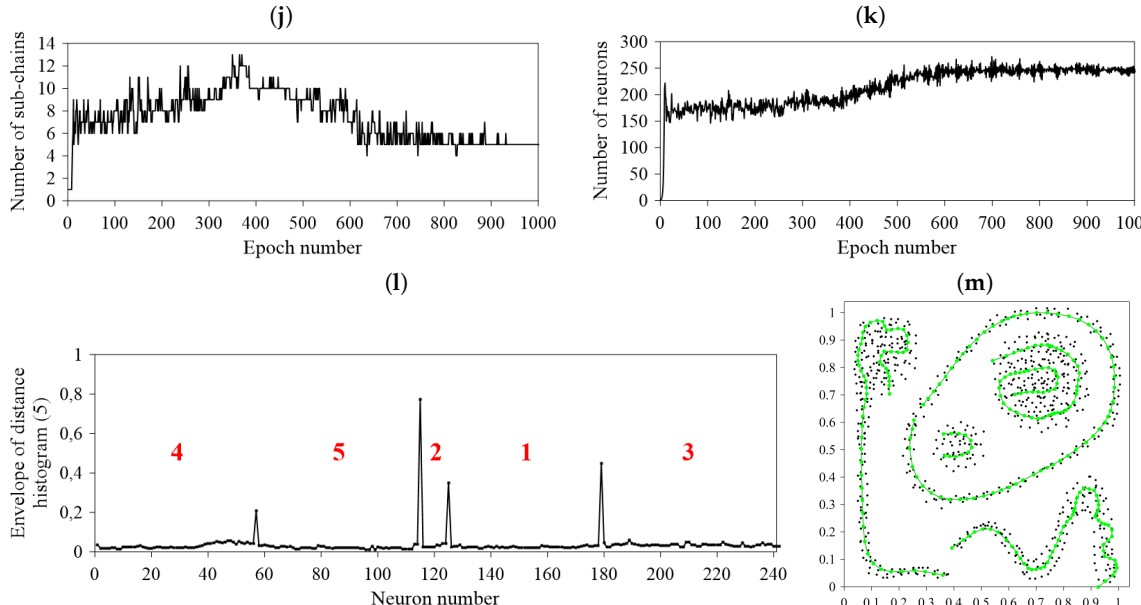

**Figure 17.** The evolution of GeSOM with 1DN (DSOM) in the synthetic data set of Figure 6a in learning epochs: (**a**) No. 0 (start of learning), (**b**) No. 5, (**c**) No. 10, (**d**) No. 20, (**e**) No. 50, (**f**) No. 100, (**g**) No. 300, (**h**) No. 500, and (**i**) No. 1000 (end of learning); moreover, plots of the number of sub-chains (**j**) and the number of neurons (**k**) vs. epoch number and (**l**) the distance histogram (5) for the network of Figure 17i as well as (**m**) the network in learning epoch No. 1000 for different starting point of the learning algorithm.

The lower the histogram's bars are, the closer the corresponding neurons are situated and thus, the data they represent belong to more compact clusters. The distance histogram illustrates the distribution of original data in particular clusters and shows distinct borders (high single bars) between clusters.

In comparison with the conventional SOM and its previously presented generalizations, our approach demonstrates an interesting and advantageous feature directly shown in Figure 17j,k. Namely, the "unfolding" stage of the network is almost eliminated. The system almost immediately increases its number of neurons to the desired level and concentrates on its essential task, i.e., detecting data clusters by appropriate distribution of neuron sub-chains in the data set.

The splitting-merging neuron chains have, however, some disadvantages regarding the multi-prototype generation for volume clusters. The corresponding neuron sub-chain that is "twisting" within the volume of the cluster, at the end of learning may assume different shapes for different starting points of the learning algorithms—compare Figure 17i with Figure 17m for clusters 1 and 2. It may result in slightly incorrect image of the data distribution in volume clusters.

The aforementioned disadvantages of GeSOMs with 1DN are eliminated in their extension in the form of generalized SOMs with splitting-merging Tree-Like Structures (GeSOMs with T-LSs) which—as their name implies—operate on tree-like networks. Figure 18a–i show the selected stages of the evolution of GeSOM with T-LSs for the synthetic data set of Figure 6a. Figure 18j–k illustrate the adjustment of the number of network substructures (finally equal to 5, i.e., the number of clusters) and the number of neurons (finally equal to 247) in the network during the learning process. This time, multi-prototypes for volume clusters—represented by tree-like structures—are much more evenly distributed (comparing with chain-based approach) within particular clusters.

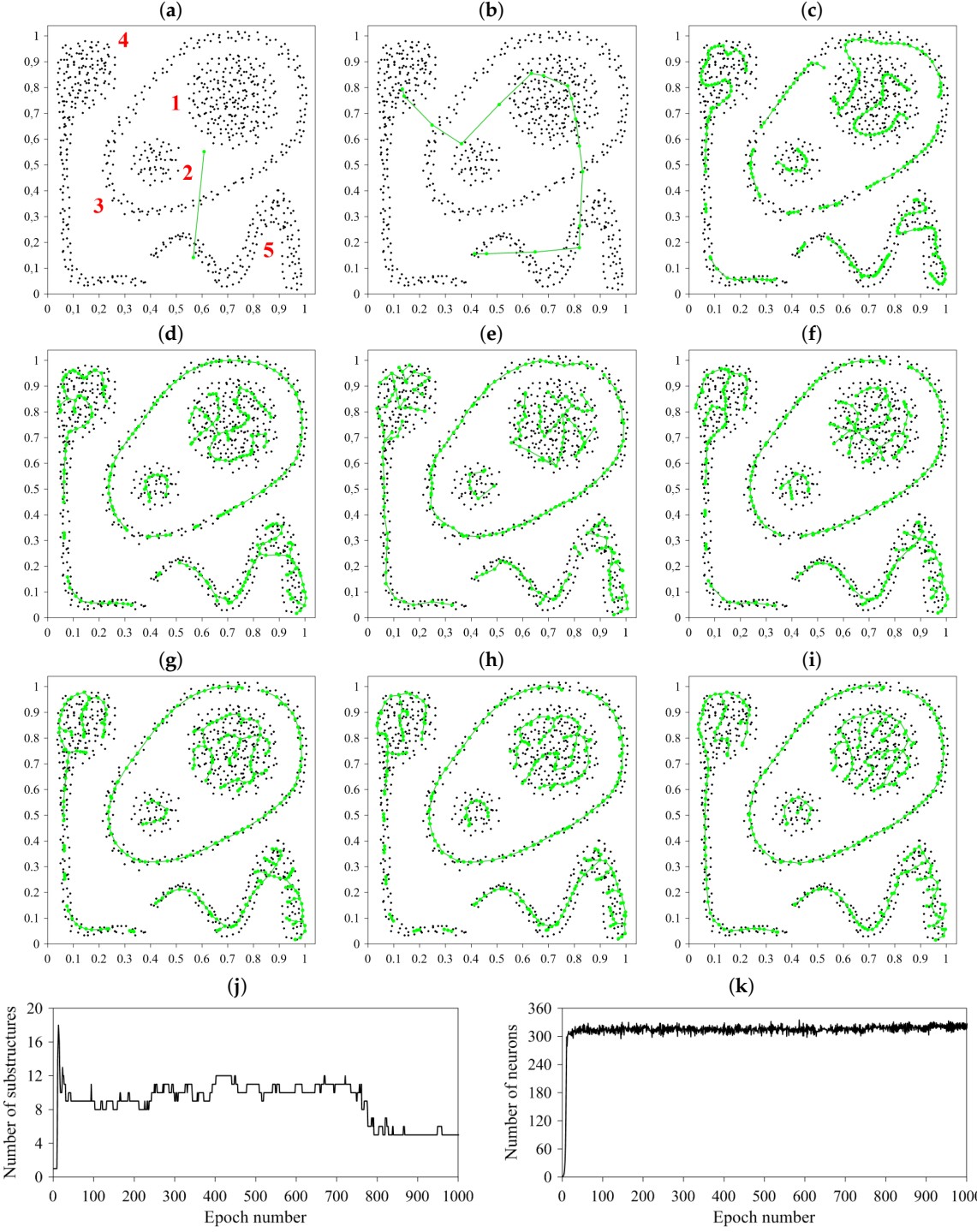

**Figure 18.** The evolution of GeSOM with T-LSs in the synthetic data set of Figure 6a in learning epochs: (**a**) No. 0 (start of learning), (**b**) No. 5, (**c**) No. 10, (**d**) No. 20, (**e**) No. 50, (**f**) No. 100, (**g**) No. 300, (**h**) No. 500, and (**i**) No. 1000 (end of learning) as well as plots of the number of substructures (**j**) and the number of neurons (**k**) vs. epoch number

Similarly as GeSOM with 1DN, also GeSOM with T-LSs almost eliminates the "unfolding" stage in its operation. Figure 18j,k show that the system almost immediately increases its number of neurons and starts discovering clusters in data. Already at the 10-th learning epoch (out of 1000)—see Figure 18c—the system develops pretty precise image of the cluster distribution in the considered data set. We can conclude that GeSOM with T-LSs is a volume-cluster-oriented version of GeSOM with 1DN.

We demonstrated that the proposed original GeSOMs with 1DN (DSOMs) and GeSOMS with T-LSs outperform different alternative approaches in various complex clustering tasks. In particular:

(a) in [13] we applied our GeSOMs with 1DN (DSOMs) to WWW-newsgroup-document clustering (the collection of 19997 documents was considered); our approach generated 58.41% of correct decisions, whereas alternative approaches achieved from 33.98% to 49.12% of correct decisions,

(b) in [11] we tested our GeSOMs with 1DN (DSOMs) in terms of their abilities to correctly determine the number of clusters in a given data set (8 benchmark data sets available from the University of California (UCI) Database Repository at https://archive.ics.uci.edu/ml were considered); our approach achieved 100% of correct decisions for 6 out of 8 considered data sets, whereas an alternative method obtained such an accuracy only for 1 data set,

(c) in [15] we applied our GeSOMs with T-LSs to microarray leukemia gene data clustering (the benchmark leukemia cancer data set containing 7129 genes and 72 samples was considered); our approach achieved 98.6% of correct decisions regarding the cluster assignments of particular data samples, whereas an alternative method gave only 93.14% accuracy,

(d) in [16] we applied our GeSOMs with T-LSs to WWW-document clustering (the collection of 548 abstracts of technical reports and its 476-element subset, both available from the WWW server of the Department of Computer Science, University of Rochester, USA at https://www.cs.rochester.edu/trs were considered); our approach obtained 87.23% and 84.87% clustering accuracies for bigger and smaller collections, respectively, whereas alternative approaches gave from 36.68% to 65.33% accuracy for bigger collection and from 38.45% to 69.96% for smaller collection,

(e) in [18] our GeSOMs with T-LSs were used to uncover informative genes from colon cancer gene expression data via multi-step clustering (the benchmark colon cancer microarray data set containing 6500 genes and 62 samples was considered); our approach generated 88.71% of correct decisions regarding the clustering of samples, whereas alternative methods achieved from 51.61% to 85.73% accuracy,

(f) in [19] we applied our GeSOMs with T-LSs to electricity consumption data clustering for load profiling (the benchmark Irish Commission for Energy Regulation data set containing 4066 customer profiles with 25728 recordings per profile was considered); our approach achieved 94.86% of correct decisions, whereas alternative methods generated from 89.77% to 94.76% of correct decisions,

(g) finally, in [17] we applied our both approaches to microarray lymphoma gene data clustering (the benchmark lymphoma cancer data set containing 4026 genes and 62 samples was considered); our approaches achieved 91.9% (GeSOMs with 1DN) and 93.6% (GeSOMs with T-LSs) of correct decisions, whereas alternative techniques gave from 61.3% to 75.8% of correct decisions.

Concluding, our approaches provide up to 20% increase in the clustering accuracy in comparison with alternative methods. It is worth emphasizing that alternative methods require the number of clusters to be predefined to perform the clustering. Thus, they are favored in regard to our approaches which aim at an automatic detection of the number of clusters and the cluster multi-point prototypes in a given data set.

There are two limitations of our approaches: (i) inability to visualize the distribution of data clusters—particularly in the case of GeSOMs with T-LSs—and this is the objective of our future work and (ii) relatively high computational complexity of both approaches (it is, however, typical for overwhelming majority of unsupervised-learning-based techniques).

## 4. Conclusions

In this paper, we briefly present several modifications and generalizations of the concept of SOMs in order to illustrate their advantages in applications that range from high-dimensional data visualization to complex data clustering. Starting from the conventional SOMs, Growing SOMs

(GSOMs), Growing Grid Networks (GGNs), Incremental Grid Growing (IGG) approach, Growing Neural Gas (GNG) method as well as our two original solutions, i.e., Generalized SOMs with 1-Dimensional Neighborhood (GeSOMs with 1DN also referred to as Dynamic SOMs (DSOMs)) and Generalized SOMs with Tree-Like Structures (GeSOMs with T-LSs) are presented. Original computer-implementations of particular solutions are developed and their detailed simulation results are shown. The performance of particular solutions is illustrated and compared by means of selected data sets. Table 1 summarizes the characteristics of the considered solutions in terms of (i) the modification mechanisms used, (ii) the range of network modifications introduced, (iii) the structure regularity, and (iv) the data-visualization/data-clustering effectiveness. The performance of particular solutions is illustrated and compared by means of selected data sets.

**Table 1.** Various modifications and generalizations of SOMs, their functionalities and data-visualization/ data-clustering effectiveness.

| Methods | SOM [1,2] | GSOM [7] | GGN [8] | IGG [9] | GNG [10] | GeSOM with 1DN [11] (DSOM [12,13]) | GeSOM with T-LSs [14–19] |
|---|---|---|---|---|---|---|---|
| **Modification mechanisms** | | | | | | | |
| Adding neurons | no | yes | yes | yes | yes | yes | yes |
| Removing neurons | no | no | no | no | yes | yes | yes |
| Adding connections | no | no | no | yes | yes | yes | yes |
| Removing connections | no | no | no | yes | yes | yes | yes |
| **Range of network modifications** | | | | | | | |
| Increasing the size | no | yes | yes | yes | yes | yes | yes |
| Reducing the size | no | no | no | no | yes | yes | yes |
| Disconnection into subnetworks | no | no | no | yes | yes | yes | yes |
| Reconnection of some of subnetworks | no | no | no | n/r[*1] | n/r[*1] | yes | yes |
| **Structure regularity** | | | | | | | |
| Fully connected | yes | yes | yes | no | no | no | no |
| Regular, rectangular structure | yes | yes | yes | yes | no | no | no |
| Form of data visualization | $U_j$ (4) | $U_j$ (4) | $U_j$ (4) | $U_j$ (4) | - | $H_j^{dist}$ (5) | - |
| **Effectiveness of** | | | | | | | |
| data visualization | ■ | ■ | ■ | ■ | — | – | — |
| data clustering | – | ■ | ■ | ■ | ■ | ■ | ■ |

[*1]n/r - not reported in the literature.

Our GeSOMs with T-LSs have been effectively used in complex clustering tasks including the clustering of huge amounts of gene expression data [15,17,18], electricity consumption data [19], and WWW documents [16]. Our future work aims at equipping our both approaches (in particular, GeSOMs with T-LSs) with data visualization mechanisms while maintaining their full capacity for automatic and precise clustering of data. We hope that such a generalization contributes to bridging two SOM-related research areas of dimensionality-reduction-based data visualization and complex data clustering.

**Author Contributions:** Conceptualization, M.B.G. and F.R.; Formal analysis, M.B.G. and F.R.; Investigation, M.B.G. and F.R.; Methodology, M.B.G. and F.R.; Writing—original draft, M.B.G. and F.R.; Writing—review & editing, M.B.G. and F.R. All authors have read and agreed to the published version of the manuscript.

**Funding:** This research received no external funding.

**Acknowledgments:** The authors would like to thank J. Piekoszewski for performing some numerical experiments for GeSOMs with T-LSs.

**Conflicts of Interest:** The authors declare no conflict of interest.

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
