# Peer review of "Evolution of SOMs’ Structure and Learning Algorithm: From Visualization of High-Dimensional Data to Clustering of Complex Data"

_algorithms, doi:10.3390/a13050109_

Round 1

Reviewer 1 Report

Authors are suggested to address the following comments.

Point 1. Abstract, please highlight the performance of proposed work numerically. Also, please highlight the percentage improvement by proposed work.

Point 2. Introduction, paragraph 1, authors mentioned “SOM is "a new, effective software tool for the visualization of high-dimensional data" (the quotation from Kohonen [1]).”, it is no longer “new” if it was proposed in 1980s.

Point 3. Introduction, paragraph 1, authors mentioned “The visualization is performed by means of a topology-preserving mapping of the considered data into a low-dimensional display space (most often, in the form of a two-dimensional, usually rectangular, grid).”, it is also for three-dimensional.

Point 4. Authors are suggested to add a paragraph to summarize the contributions of this paper, preferably in point-form.

Point 5. It is suggested authors to group section 2 to section 5 into single section which are basically background information and literature review.

Point 6. Literature review focusing mainly on recently published articles (2015-2020 and preferably journal articles) is missing. It is also expected to summarize the performance and inadequacies of existing works.

Point 7. This paper is lack of proper explanation with the aid of formulation/equations.

Point 8. There are many plots and results which are nice having. However, it is expected thorough explanation should be presented. Particularly, when it comes to many subfigures, authors should explain clearly and guide readers how to read and interpret the findings. Please apply throughout the manuscript. It is expected a significantly elaboration is made in revision.

Point 9. Performance comparison between proposed work and related works (2015-2020) is expected.

Point 10. Please discuss the limitations of proposed work.

Point 11. Please discuss future work.

Author Response

Please see the attached file "Gorzalczany_Rudzinski_Cover letter_Response to reviewers comments.pdf", which contains:
1. Cover letter.
2. Response to Reviewer 1 Comments
3. Response to Reviewer 2 Comments
4. Special copy of our revised paper with clearly highlighted (using red font) revisions.

Reviewer 2 Report

To the Authors:

Hereby, After the reviewing process, I sending my comments of the paper: “Evolution of SOMs’ structure and learning algorithm -from visualization of high-dimensional data to clustering of complex data”.

The authors stated that they performed several modifications self-organizing neural networks with Generalized SOMs with 1-Dimensional Neighborhood (GeSOMs) and Generalized SOMs with Tree-Like Structures. However, there is no deep and extensive explanation about, why both methods could improve the process for visualization of high-dimensionality data to clustering complex data. The best figures (15,16, 17 & 18) that could explain their research work, there is no substantial explanation about the figures. And Figure 18 is not even cited in the paper.

The abstract section is missing a one important part to quantify the obtained results. The structure of the work is not following the journal structure format, the paper has pictures and tables in the conclusions part.

Finally, the paper has to go through an extensive English correction. The authors paid little attention to orthographic details.

Author Response

(The authors gave the same response as above.)

Round 2

Reviewer 1 Report

Authors have thoroughly addressed the comments and improved the quality of the manuscript. I have two follow-up comments.

Follow-up comment 1: Authors are suggested to rewrite the contribution. Normally, it is (i) percentage improvement on some indicators (e.g. accuracy, time complexity) compared with existing works; (ii) proposed work addresses the limitations of existing works; (iii) some discussion has been firstly studied and presented.

Follow-up comment 2: In some figures, E.g. Figure 8, in some subfigures, please correct “,” with “.” for numerical values.